# Pentoxifylline in the Treatment of Cutaneous Leishmaniasis: A Randomized Clinical Trial in Colombia

**DOI:** 10.3390/pathogens11030378

**Published:** 2022-03-21

**Authors:** Maria del Mar Castro, Alexandra Cossio, Adriana Navas, Olga Fernandez, Liliana Valderrama, Lyda Cuervo-Pardo, Ricardo Marquez-Oñate, María Adelaida Gómez, Nancy Gore Saravia

**Affiliations:** 1Centro Internacional de Entrenamiento e Investigaciones Médicas (CIDEIM), Cali 760031, Colombia; acossio@cideim.org.co (A.C.); adriana.navas@radboudumc.nl (A.N.); ofernandez@cideim.org.co (O.F.); lilianavalderrama@yahoo.com (L.V.); lyda.cuervopardo@medicine.ufl.edu (L.C.-P.); ricardomarquezonate@gmail.com (R.M.-O.); mgomez@cideim.org.co (M.A.G.); 2Universidad Icesi, Cali 760031, Colombia

**Keywords:** cutaneous leishmaniasis, pentoxifylline, meglumine antimoniate, clinical trial

## Abstract

Addition of the immunomodulator pentoxifylline (PTX) to antimonial treatment of mucosal leishmaniasis has shown increased efficacy. This randomized, double-blind, placebo-controlled trial evaluated whether addition of pentoxifylline to meglumine antimoniate (MA) treatment improves therapeutic response in cutaneous leishmaniasis (CL) patients. Seventy-three patients aged 18–65 years, having multiple lesions or a single lesion ≥ 3 cm were randomized to receive: intramuscular MA (20 mg/kg/day × 20 days) plus oral PTX 400 mg thrice daily (intervention arm, *n* = 36) or MA plus placebo (control arm, *n* = 37), between 2012 and 2015. Inflammatory gene expression was evaluated by RT-qPCR in peripheral blood mononuclear cells from trial patients, before and after treatment. Intention-to-treat failure rate was 35% for intervention vs. 25% for control (OR: 0.61, 95% CI: 0.21–1.71). Per-protocol failure rate was 32% for PTX, and 24% for placebo (OR: 0.50, 95% CI: 0.13–1.97). No differences in frequency or severity of adverse events were found (PTX = 142 vs. placebo = 140). Expression of inflammatory mediators was unaltered by addition of PTX to MA. However, therapeutic failure was associated with significant overexpression of *il1β* and *ptgs2* (*p* < 0.05), irrespective of study group. No clinical benefit of addition of PTX to standard treatment was detected in early mild to moderate CL caused by *Leishmania* (*V*.) *panamensis*.

## 1. Introduction

An estimated 600,000 to 1 million new cases of cutaneous leishmaniasis occur worldwide annually [1]. Colombia is among the ten countries globally reporting the highest number of cases [1]. Efficacy of toxic first-line pentavalent antimonial drugs in the treatment of cutaneous leishmaniasis (CL) varies widely [2,3,4], being as low as 25% in pediatric populations [5], and contraindicated in pregnancy and individuals having co-morbidities [6,7]. The outcome of infection by dermatotrophic *Leishmania* species is intimately linked to the inflammatory response elicited by the infecting parasite [8]. Furthermore, the outcome of treatment is influenced by the interplay of the host immune and inflammatory response and both the antimicrobial and modulatory effects of therapeutic agents on drug transport and the inflammatory response. Co-adjuvant approaches to treatment of CL seek to increase efficacy and reduce exposure to toxic antileishmanial drugs exemplified by first line pentavalent antimonials.

Pentoxifylline, which inhibits the production of TNF-α and other proinflammatory mediators [9,10], has been used in combination with pentavalent antimonials to modulate the chronic inflammatory response associated with mucosal leishmaniasis caused by *Leishmania (V.) braziliensis* [11,12,13]. Oral pentoxifylline, together with meglumine antimoniate (MA), a pentavalent antimonial, resulted in the resolution of 9/10 cases of mucosal leishmaniasis that had not responded to prior antimonial therapy [11]. A subsequent randomized study [12] reaffirmed the significant therapeutic benefit of this combination, achieving clinical resolution of mucosal lesions with a single cycle of treatment (100% MA + pentoxifylline vs 59% with MA + placebo), and promoting more rapid healing than antimony alone.

A pilot randomized placebo controlled clinical trial in Brazil [14] did not yield a significant difference, but rather a trend interpreted as suggesting that addition of pentoxifylline might improve efficacy of pentavalent antimonial therapy for CL. An immunomodulatory effect, principally mediated by decreased production of TNF-α, was proposed as the mechanism of co-adjuvant effects in the treatment of dermal leishmaniasis [12,14,15]. Later, a larger randomized, placebo-controlled trial of the combined administration of pentoxifylline and meglumine antimoniate for the treatment of CL caused by *L. (V.) braziliensis* found no difference in either cure rate or time to healing, and a higher frequency of adverse effects in the group receiving pentoxifylline [16]. 

The variability of therapeutic response to the addition of pentoxifylline to treatment with pentavalent antimonial drugs, together with the unmet need for more effective, less toxic regimens for the most frequent presentation of CL, motivated us to conduct this study. In this randomized placebo-controlled double-blind trial, we assessed the efficacy and safety of this combination therapy for localized, uncomplicated CL in Colombia, where *Leishmania (V.) panamensis* is the predominant species. We also report changes in inflammatory gene expression in peripheral blood mononuclear cells from trial patients, before and after treatment. 

## 2. Results

### 2.1. Participant Recruitment and Clinical Presentation of CL 

Patients were enrolled from February 2012 to June 2014 (last follow-up visit completed in December 2014). Eighty-eight patients were assessed for eligibility; fifteen did not meet the inclusion criteria. Seventy-three participants were enrolled: 36 in the intervention (pentoxifylline + MA-PTX + MA-) and 37 in the control (placebo + MA-placebo + MA-) arm (Figure 1). The low frequency of eligible patients (most consulting patients presented single lesions of less than 3 cm in diameter) during the approved timeframe of the project limited enrollment. Therefore, interim analysis and review by the data safety committee were undertaken before completion of the calculated sample size (*n* = 100). Based on the outcome of the review, the estimated duration of further recruitment needed to reach the sample size, and subsequent unbudgeted additional costs of continuing this Good Clinical Practices (GCP) compliant trial, enrollment was stopped. Three patients were excluded post-randomization: two did not receive study interventions (did not attend any visit after enrollment and randomization), and one was a protocol deviation due to incorrect inclusion in the study (single lesion of less than 3 cm, detected by the study coordinator). Four patients were lost to follow-up (Figure 1). 

Sociodemographic and clinical characteristics were similar for the two study arms (Table 1). Participants were predominantly Afro-Colombian males; the median age was 28 and 36 years in the experimental and control arm, respectively. Most of the patients had 1–3 lesions and a short duration of disease (median 1 and 2 months for PTX + MA and placebo + MA, respectively). Parasites were isolated and identified in 78.6% of participants (55/70); with *L. (V.) panamensis* being the predominant species (90.9%; 50/55) (Table 1). 

Clinical presentation of CL of enrolled patients was generally mild. Only 11.76% (4/34) of the patients in the pentoxifylline arm and 8.33% (3/36) in the control arm presented with moderate CL (defined as having more than three lesions, and at least one of the lesions having a diameter > 5 cm). Most patients had ulcerated lesions (94% in PTX vs. 91.7% in placebo arm having at least one ulcerated lesion), and less than 33.3% of participants presented abnormalities in the lymph nodes or in lymphatic ducts. Most of the lesions were located in the arms (54.8% and 50% in the intervention and control arm, respectively, Table 1). 

### 2.2. Efficacy

There was no evidence of effect on the therapeutic response of CL patients with the addition of PTX to the standard treatment with MA in either the intention-to-treat (ITT) or per-protocol (PP) analyses. In the ITT analysis, definitive cure (defined as complete re-epithelization, absence of inflammatory signs for all cutaneous lesions, and absence of new leishmaniasis lesions at 26 weeks after initiation of treatment [17]) was 64.7% (22/34) in the PTX + MA arm and 75% (27/36) in the placebo + MA arm (OR = 0.61, 95% CI: 0.21–1.71) (Table 2). In this ITT analysis, four patients who were lost to follow-up (three in PTX + MA arm, and one in placebo + MA) were considered as treatment failures, regardless of the trial arm [18]. Results of the PP analysis followed a similar trend, with 70% cured (14/20) in the PTX + MA arm and 82.1% cured (23/28) in the placebo + MA arm (OR = 0.50, 95% CI: 0.13–1.97) (Table 2); as well as the results of a sensitivity analysis with a complete case approach [18] for the ITT: 70.9% cured (22/31) in the PTX + MA arm and 77.14% cured (27/35) in the placebo + MA arm (OR = 0.72, 95% CI: 0.23–2.18).

As a secondary outcome, we found that addition of PTX to CL treatment with MA did not significantly change the cure rates at the end of treatment on day 21, or weeks 5, 7, 13 and 26 post-treatment; *p* > 0.07 for all the comparisons (Appendix A).

### 2.3. Safety

Three participants presented serious adverse events (AEs); two in the intervention arm and one in the placebo arm. These were not associated with the study interventions: one was due to a gunshot wound, and two had sharp weapon injuries.

Frequency of non-serious adverse events was similar in both arms: 142 in the intervention and 140 in the control arm, with a median (range) of 3 (1–13) and 3 (1–11), respectively, for the intervention and control arms (*p* = 0.77). Most frequent AEs were fever, headache, increased levels of aspartate aminotransferase (AST), alanine aminotransferase (ALT) and amylase, arthralgia and reaction at the injection site. Only six patients did not report AEs, two in the PTX and four in the placebo arm (Table 3). 

Clinical and laboratory AEs were predominantly mild. In total, 24/282 (8.5%) adverse events were classified as grade 2, and 3/282 (1%) as grade 3. Headache, increased levels of liver enzymes or of amylase represented most of the moderate AEs, and all subsided during follow-up post-treatment. 

Adverse events did not lead to a medical decision for the suspension of the study drug in any of the treatment arms. However, some patients abandoned treatment due to AEs (11.7% (4/34) in the intervention arm; 11.1% (4/36) in the control arm). Stated reasons for treatment dropout were myalgia, arthralgia, reaction at the injection site, fever, dizziness, headache and abdominal pain.

### 2.4. Compliance with Treatment and Study Protocol

Losses to follow-up were similar in both arms, at 8.8% (3/34) in the pentoxifylline and 2.8% (1/36) in the placebo arm (Figure 1). We found no differences regarding clinical characteristics or frequency of adverse events in the patients who were lost to follow-up, compared to the patients who completed the study visits.

Information on compliance with treatment was recorded in 98.57% (69/70) patients. Overall adherence to the prescribed treatment was similar between the trial arms: 64.7% (22/34) patients in the intervention arm received ≥ 90% of the prescribed dose of PTX + MA, and 80% (28/35) of patients in the control arm received ≥ 90% of placebo + MA (*p* = 0.15).

However, assessment of each specific drug showed that adherence to PTX was lower (70.6%; 24/34) than adherence to placebo (94.3%; 33/35) (*p* = 0.009), and adherence to MA alone was 76.5% in the intervention and 82.9% in the control arm.

Among participants lost to follow-up, compliance with treatment in the PTX arm was >90% in two patients and <90% in one. Compliance of the only participant lost in the placebo arm was unknown.

### 2.5. The Immunomodulatory Activity of PTX Ex Vivo Involves Changes in Gene Expression of Pro and Anti-Inflammatory Mediators

Modulation of inflammatory pathways, alternative to TNF-α, has been recognized as part of the systemic activity of PTX [19]. However, how engagement of these immune pathways impacts the overall therapeutic effect of PTX remains poorly understood. To approach this knowledge gap, we explored the effect of PTX on the gene expression of a set of pro- and anti-inflammatory mediators and receptors in mononuclear cells from CL patients participating in the clinical trial. The expression of a panel of 84 genes coding for immune mediators and receptors was screened ex vivo in PBMCs (peripheral blood mononuclear cells) obtained before treatment from three enrolled CL patients and then exposed ex vivo to PTX. Of these, expression of 22 genes were not detected (Appendix A).

As shown in Figure 2 (and Appendix A), ex vivo exposure of uninfected PBMCs to PTX significantly induced the expression of 14 genes linked to the activation and chemoattraction of monocytes and neutrophils when compared to the untreated PBMCs (*ccl3*, *ccl4*, *ccl7*, *cxcl1*, *cxcl2*, *cxcl3*, *cxcl5*, *il1a*, *il1b*, *il1r1*, *il6*, *il8*, *ptgs2* and *tlr2*). In addition, 13 genes were downregulated and enriched in receptors involved in innate cell signal transduction (*c3*, *ccl24*, *ccr2*, *ccr3*, *cd40*, *csf1*, *il10*, *tlr4*, *tlr5* and *tlr7*), and molecules involved in Th1 cell responses (*cxcl10*, *cxcl9*, *fasl*). The other 35 genes were not significantly modulated.

### 2.6. There Was No Evidence That Addition of PTX to the First-Line Treatment with MA Differentially Modulated the Expression of the Inflammatory Mediators by PBMCs Evaluated at the End of the Treatment

Six inflammatory genes were selected for evaluation in PBMCs from participants of the clinical trial: *cxcl10*, *ccl2*, and *cfs1* as representatives of genes downregulated by PTX, and *cxcl5*, *il-1β* and *ptgs2* as representative of upregulated genes. These six genes were consistently modulated by PTX in both infected and uninfected PBMCs (Appendix A).

The expression of the selected genes was evaluated in PBMCs collected prior to initiation and at end of treatment, in a subset of 11 participants in the intervention arm (MA + PTX) and 11 participants in the control arm (MA + placebo). The ratio between gene expression at the beginning and the end of treatment was compared between groups. No differences were found in the expression of these mediators between study arm groups (Figure 3). However, analysis of gene expression in relation with the outcome of treatment (cure and failure independently of the treatment received), showed a significant overexpression of *il1β* and *ptgs2* in patients presenting therapeutic failure (Figure 4).

## 3. Discussion

Addition of pentoxifylline to the standard treatment with pentavalent antimony did not increase its efficacy in our population of patients with mild to moderate cutaneous leishmaniasis, caused by *L. (V.) panamensis*. This conclusion is based on the comparison of proportion of cure at six months, and at different follow-up times, between patients receiving combination treatment with pentoxifylline or placebo (intention-to-treat OR: 0.61, 95% CI: 0.21–1.71; and per-Protocol OR: 0.50, 95% CI: 0.13–1.97). Our results are similar to previous reports from Brazil [14,16] using this combination treatment for CL patients predominantly infected with *L. (V.) braziliensis* having mild to moderate clinical presentation. Together, these findings evidence the lack of therapeutic gain by addition of pentoxifylline to the standard antimonial treatment for American CL caused by *L. (V.) panamensis* and *L (V.) braziliensis,* the species most prevalent among CL patients in Central and South America.

Immunomodulation has been proposed as a host-targeted approach for the treatment of CL, considering the role of the inflammatory response and its regulation in the pathogenesis and therapeutic outcome of *Leishmania* infection [20,21]. The modulation of IFN-γ and TNF-α production by pentoxifylline [15,22] has been proposed as a strategy for improving the therapeutic response in mucosal leishmaniasis [12,23]. This, combined with the reported safety profile and low cost of pentoxifylline, has made it an attractive option for combination therapy. Our ex vivo experiments confirm that the effect of exposure to PTX extends beyond modulation of IFN-γ and TNF-α, revealing gene expression signatures consistent with activation and recruitment of monocytes and neutrophils and repression of Th1 responses. Six representative members of this gene repertoire were evaluated in samples collected after in vivo drug exposure, from subgroups of study participants in the intervention and control arms, and no difference was found in their expression. Absence of difference in expression of immune-inflammatory genes (*cxcl10*, *cxcl5*, *ccl2*, *il1b*, *ptgs2* and *cfs1*) in PBMCs from patients in the PTX and placebo arms of the present study concur with previous findings of Brito and collaborators [14], in relation with secretion of CXCL10, CXLC9, CCL3 and IL-10 observed in the culture supernatants of PBMC (re-stimulated ex vivo with soluble *Leishmania* antigen) from CL patients who received antimony plus pentoxifylline and CL patients receiving antimony plus placebo [14]. In the latter study, statistically significant differences in the secretion of cytokines between arms were observed only for TNF-α and IFN-γ, with a more pronounced decrease of these cytokines in the antimony plus pentoxifylline group, without evidence of clinical benefit. Differences observed in the modulation of cytokines and chemokines ex vivo and in vivo may be explained by PTX metabolism and pharmacokinetics in vivo [24,25].

The aforementioned results concur with the absence of clinical benefit in treatment outcome. However, analysis of gene expression of these mediators in relation with the therapeutic response showed that *il1β* and *ptgs2* were significantly increased in PBMCs from patients who failed treatment. This suggests that, regardless of the therapeutic scheme received, higher levels of expression of pro-inflammatory mediators including *IL-1β*, or lack of regulation of their expression during the course of treatment, may contribute to therapeutic failure.

Small clinical studies including a randomized trial of co-adjuvant use of pentoxifylline in patients with mucosal leishmaniasis caused by *L. (V.) braziliensis* have provided evidence of benefit of this combination [11,12]. Notably, when evaluated for treating mild to moderate cutaneous leishmaniasis caused by *L. (V.) braziliensis* and now *L. (V.) panamensis*, these findings were not replicated in two trials for American CL by Brito et al. [14,16] and this study, which found no therapeutic gain in adding pentoxifylline to MA. The different outcomes in CL and ML could be partially explained by the pathogenesis of ML, which is associated with hyper-reactive cell-mediated inflammatory responsiveness to *Leishmania* antigen and low, persistent parasite burden, features that contribute to chronic inflammation and tissue destruction [26]. Duration of treatment is 50% longer for ML (daily administration of 20 mg Sb^V^/kg for 30 days) than CL patients (20 days daily administration) [6]. The dose regimen of 1200 mg/day is the most commonly used for pentoxifylline combination therapy for different dermatologic indications [22]. Our patients and those in the Brazilian CL trials received this daily dose during the 20 days of antimonial treatment, which is shorter in comparison to recommended ML treatment (30 days) and other dermatologic indications for pentoxifylline, such as venous ulcers of the leg (6 to 24 weeks of treatment) [27]. An alternative combination treatment regimen to increase exposure to PTX could be explored; however, extending the length of treatment and the number of doses per day may affect adherence [28] because of adverse effects, as well as the additional days of therapy.

Pentoxifylline did not modify the safety profile of standard antimonial treatment in the study population. Reported clinical and laboratory adverse events in our patients were all among those expected for antimonial treatment [29], which is consistent with previous studies assessing the combination of MA and PTX [12,14,16,30]. Dizziness was more frequent in the pentoxifylline arm in this study, but the difference was not significant. Notably, patients receiving pentoxifylline were less compliant with tablets than the placebo control arm (*p* = 0.009), which suggests a potential effect of additional AEs on adherence to treatment.

We did not reach the estimated sample size, which is a limitation of the study. Recruitment was limited by the lower-than-expected frequency of the eligible clinical presentations and logistic and social challenges to conduct clinical trials in remote rural areas, where approximately 80% of patients with CL in Colombia live and reside [31]. This and other trials conducted within the context of endemic transmission of CL also underscore the challenges of fulfillment of regulatory requirements designed for commercially sponsored trials by institutional sponsors of non-commercial public health driven clinical trials [32] that are vital to the development of therapeutics for NTDs.

We have not presented post-hoc power calculations because point estimates and confidence intervals are more informative [33]. The confidence intervals for the intention-to-treat (ITT) and per-protocol (PP) analyses of this study are wide and include the null value. Nevertheless, our findings are consistent with the recent evidence on the co-adjuvant use of pentoxifylline for New World CL caused by *Leishmania (V.) braziliensis* [16], supporting the lack of therapeutic gain of this combination. Another potential limitation is having three post-randomization exclusions. One of these exclusions was due to a recruitment error (one patient did not meet inclusion criteria), and two patients were lost before initiating treatment. None of these exclusions (two participants in the intervention, one in the control arm) were related to patient outcomes or differences after randomization. Therefore, the risk of bias for excluding these patients is low [34].

This study provides further evidence that the use of pentoxifylline in combination with meglumine antimoniate does not improve the clinical response to antimonial drugs in mild to moderate presentations of cutaneous leishmaniasis caused by *L. Viannia* species. Combination therapies are considered in the target product profiles for CL, and mitigation of the host inflammatory response is a rational approach for treatment of cutaneous leishmaniasis. In conclusion, the absence of significant improvement in therapeutic response in uncomplicated mild to moderate clinical presentations of CL in this and prior Brazilian studies precludes the use of PTX in the increasingly predominant early, mild presentations of CL, as national and regional plans for elimination drive opportune case detection.

## 4. Materials and Methods

### 4.1. Study Design and Participants

We conducted a randomized (1:1), double-blinded (investigator and participant), placebo-controlled, parallel arm clinical trial to assess the therapeutic gain of pentoxifylline in the clinical and immunologic response to meglumine antimoniate (MA) treatment for cutaneous leishmaniasis (CL). Reporting of the trial was prepared following the CONSORT guidelines [35]. Patients were enrolled from February 2012 to May 2014 in the clinical facilities of *Centro Internacional de Entrenamiento e Investigaciones Médicas* (CIDEIM), in the cities of Cali and Tumaco, Colombia.

Inclusion criteria for patients were: age between 18 and 65 years; parasitological confirmation of CL; duration of disease ≥ 1 month; and either multiple lesions (>1) or a single lesion of at least 3 cm on its longest axis, presentations associated with a robust immune inflammatory response that might benefit from the immunomodulatory capacity of pentoxifylline. Exclusion criteria were: positive pregnancy test (urine); mucocutaneous disease; medical history of cardiac, renal or hepatic disease; use of any antileishmanial drug during the three months prior to enrollment; HIV positive test, and baseline values for hemoglobin, amylase, aspartate aminotransferase (AST), alanine aminotransferase (ALT), creatinine, or serum urea nitrogen (BUN) outside the normal range. In case of borderline values, decision for inclusion was supported by clinical assessment. Contraception (Depo-provera^®^) was administered to premenopausal women prior to treatment.

Two exclusion criteria were modified by protocol amendment during the trial: positive HTLV-1 (human T-lymphotropic virus type 1) test and abnormal electrocardiogram. For the latter, we specified that only patients with abnormalities in cardiac rhythm or electric conduct (bundle branch block, A-V block) were excluded. HTLV-1 positive test was discontinued as an exclusion criterion, due to the limited access to confirmatory tests and care options for patients testing positive, which will create an unnecessary burden to the participants. These amendments to the protocol were approved by the institutional ethics committee of CIDEIM.

Patients who were not eligible for the study or declined to participate received standard-of-care treatment in accordance with Colombian Ministry of Health guidelines [36].

### 4.2. Randomization

A balanced block (*n* = 6) randomization scheme that was unknown to the investigators was used. Allocation concealment was performed using the in-house software LydaR at the time of enrollment.

### 4.3. Study Interventions

Participants received meglumine antimoniate (Glucantime^®^ Sanofi-Aventis, Paris, France) intramuscular injections (IM) at a dose of 20 mg/Kg daily for 20 days, plus either pentoxifylline 400 mg orally three times a day or placebo three times a day for 20 days. Placebo and pentoxifylline tablets, provided by the pharmaceutical company Tecnoquímicas (Cali, Colombia), had similar appearance (shape, color and blister packaging). Treatment was administered at CIDEIM facilities (Cali and Tumaco, Colombia) by trained study personnel, or by trained health volunteers in some rural areas of Tumaco.

### 4.4. Clinical and Laboratory Procedures and Follow-Up

Clinical evaluation was conducted at enrollment, end of treatment, and at 7, 13 and 26 weeks after initiation of treatment (+/−7 days). All lesions were measured using standard digital calipers and photographed during each visit. The criteria defining moderate CL were: presentation of more than three lesions, and at least one of the lesions having a diameter ≥ 5 cm. *Leishmania* species were identified using a panel of previously described and validated monoclonal antibodies [37,38,39] (*L. (Viannia)* subgenus: B-2; *L. (V.) panamensis*/*L. (V.) braziliensis* species: B-12; *L. (V.) panamensis*/*L. (V.) guyanensis*: B-8, B-21; *L. (V.) panamensis*: B-4, B-11; *L. (V.) braziliensis*: B-16, B-18; *L. (V.) guyanensis*: B-19; *L. (L.) donovani*: D-2; *L. (L.) mexicana*/*L. (L.) amazonensis*: M7) or isoenzyme electrophoresis, as previously described [40].

Electrocardiogram, hemoglobin, amylase, AST, ALT, creatinine, and BUN were measured at the end of treatment (EoT) to monitor potential drug-related toxicity. Patients presenting with abnormal laboratory values were monitored until they normalized. For analysis of immunomodulation, cryopreserved PBMCs from a sample (20 mL) of venous blood obtained before and at the end of treatment were utilized.

At each follow-up visit we assessed clinical and laboratory adverse events (AEs), and measured adherence to treatment at EoT. All AEs were graded and reported according to the CTCAE [41]. Patients completed a drug diary during treatment and returned the blinded blister packs of pentoxifylline/placebo to verify compliance.

In case of therapeutic failure, rescue treatment was prescribed in accordance with Colombian National Guidelines [36] and was monitored by clinical personnel of the Leishmaniasis Program of CIDEIM.

### 4.5. Outcome Definitions

The primary outcomes were: (1) definite cure, defined as complete re-epithelization, absence of inflammatory signs for all cutaneous lesions, and absence of new leishmaniasis lesions at 26 weeks after initiation of treatment [17]; and (2) frequency and severity of adverse events. Secondary outcomes included: (1) time-to-cure of cutaneous lesions (measured at all follow-up visits after EoT); and (2) changes in gene expression of inflammatory markers (*cxcl10*, *cxcl5*, *ccl2*, *il1b*, *ptgs2* and *csf1*).

Determination of the primary outcome was performed by three observers, including the study physician at the study site where the case presented, and review of photographs of lesions by two independent physicians of the CIDEIM Clinical Unit.

### 4.6. Immune Response

To probe the immunomodulatory context during treatment of CL patients in the placebo and pentoxifylline arms of the clinical trial, we first explored the ex vivo effect of PTX on the expression of a set of 84 pro- and anti-inflammatory mediators and receptors (gene list provided in Appendix A) in PBMCs obtained from three patients with CL before treatment within the study arms. The analysis of these 84 genes, and the relationship of gene products with the therapeutic outcome of CL identified in previous studies (*ccl2*, *cxcl10*, *ptgs2* and *csf1*) [14,42], informed the selection of the six relevant mediators (*cxcl10*, *cxcl5*, *ccl2*, *il1b*, *ptgs2* and *csf1*) for the comparison of trial arms in a convenience sample of 11 participants per arm. Gene expression by individual patient PBMCs was quantitatively evaluated before initiating treatment, and at the end of treatment on day 20 in the presence or absence of infection ex vivo with *L. (V.) panamensis.* Gene expression assays were conducted with TaqMan probes (Applied Biosystems) for *cxcl10* (Hs00171042_m1), *cxcl5* (Hs00171085_m1), *ccl2* (Hs 00234140_m1), *il1b* (Hs01555410_m1), *ptgs2* (Hs00153133_m1) and *csf1* (Hs00174164_m1), analyzed by absolute quantitation based on extrapolation using a standard curve, and expressed as a ratio in relation with GAPDH (Hs99999905_m1).

### 4.7. Leishmania Strains, Peripheral Blood Cell Isolation and Infection

Promastigotes of the *L. (V.) panamensis* strain susceptible to MA (MHOM/CO/2002/3594), stably transfected with the luciferase reporter gene (L.p.-LUC001) [43], were cultured at 25 °C in RPMI 1640 (22400105; Gibco, New York, NY, USA) supplemented with 10% heat-inactivated fetal bovine serum (HIFBS, 10082; Gibco), 1% penicillin-streptomycin solution (100 µg/mL streptomycin and 100 U/mL penicillin; Gibco BRL, New York, NY, USA) and 120 µg/mL geneticin G418 (108321-42-2; Sigma-Aldrich, St. Louis, MO, USA). PBMCs were isolated by centrifugation over a Ficoll-Hypaque 1077 gradient (SD10771A; Sigma, St. Louis, MO, USA). Cells were cryopreserved in 10% DMSO and 90% FBS. For experiments, cells were thawed and resuspended in complete RPMI (10% FBS, 100 µg/mL streptomycin and 100 U/mL penicillin). Cell viability was determined with Trypan Blue (Sigma) and defined as acceptable when above 80%. PBMCs were infected with stationary phase L.p.-LUC001 promastigotes at a 1:10 parasite-to-monocyte ratio for 24 h at 34 °C and 5% CO_2_ prior to evaluation of gene expression experiments. The ex vivo effect of PTX (P1784; Sigma) was evaluated at 200 uM, together with 32 ug Sb^V^/mL for MA (Walter Reed 214975AK; lot no. BLO9186 90-278-1A1 W601; antimony analysis, 25–26.5% by weight), a concentration approximating the *C*_max_ of Sb^V^ during treatment [44].

### 4.8. Gene Expression

For the initial exploratory experiments, the expression of 84 inflammatory mediators and receptors including chemokines, cytokines and associated receptors was measured in PBMCs of three patients before initiating treatment, by real-time quantitative PCR (RT-qPCR) (PCR Arrays, Cat # PAHS-077ZD, Qiagen, Hilden, Germany). Total RNA was extracted from PBMCs using Trizol reagent (Invitrogen, Carlsbad, CA, USA), followed by RNA cleanup with a RNeasy Mini Kit Columns (Qiagen, Germantown, MD, USA). cDNA was synthesized using the RT first strand synthesis kit (Qiagen). RT-qPCR reactions were run on a CFX96 Real Time System (Bio-Rad, Marnes-la-Coquette, France). Data was normalized using five housekeeping genes: β-actin (ACTB), glyceraldehyde-3-phosphate dehydrogenase (GAPDH), hypoxanthine guanine phosphoribosyl transferase (HPRT1), β-2-microglobulin (B2M), and ribosomal protein large P0 (RPLP0). Fold change of gene expression was calculated by the ΔΔCt method.

To determine whether there were differences in the modulation of the expression of the inflammatory mediators between the trial groups, an index (ratio) of the magnitude of the change in the expression of the post-treatment mediator (visit 2, V2) was calculated in relation to the magnitude of the change in the expression of the same mediator in the pre-treatment visit (visit 1, V1). The formula to calculate this value is described below:(1)$$\mathrm{Ratio}=\frac{\mathrm{Fold}\;\mathrm{Change}\;V2\;\mathrm{post}-\mathrm{treatment}\;\left(2^{-\Delta \Delta \mathrm{Ct}}\;Leishmania\;\mathrm{infection}\;\mathrm{vs}\;\mathrm{unstimulated}\right)}{\mathrm{Fold}\;\mathrm{Change}\;V1\;\mathrm{pre}-\mathrm{treatment}\;\left(2^{-\Delta \Delta \mathrm{Ct}}\;Leishmania\;\mathrm{infection}\;\mathrm{vs}\;\mathrm{unstimulated}\right)}$$

Values equal to 1 suggest of the absence of modulation of gene expression, values > 1 indicate an over-expression of the mediator, and values < 1 indicate an inhibition of gene expression at the end of treatment.

### 4.9. Sample Size and Statistical Analysis

The planned sample size of 100 patients (50 individuals per arm) was calculated to ascertain at least 25% difference in clinical outcome between the arms, with 80% power and alpha = 0.05 (two-sided), and loss of 15% of patients to follow-up. These estimates considered previous reported differences in proportion of cure after addition of pentoxifylline to antimonial treatment of mucosal and cutaneous patients, respectively, in Brazil (59%) [12] and Iran (30%) [30].

Data were verified by double entry prior to analysis. The t-test or Mann–Whitney test were used to compare quantitative data, according to their distribution. Differences in proportions were estimated with χ^2^ or Fisher’s exact test (if expected frequency of the corresponding cell < 5). The odds ratio and proportion of definitive cure by treatment arm (primary outcome) were estimated. We also estimated the frequency of adverse events per treatment group, and compared them by severity, intensity and their relationship with the study intervention. Differences in proportions of cure at weeks 5, 7, 13 and 26 (secondary outcome) were calculated.

All analyses were performed by intention-to-treat and per-protocol, using Stata^®^-12; *p* < 0.05 was considered significant. PP analysis included patients who completed follow-up visits and received ≥ 90% of the prescribed dose of MA and PTX or placebo.

## Figures and Tables

**Figure 1 pathogens-11-00378-f001:**
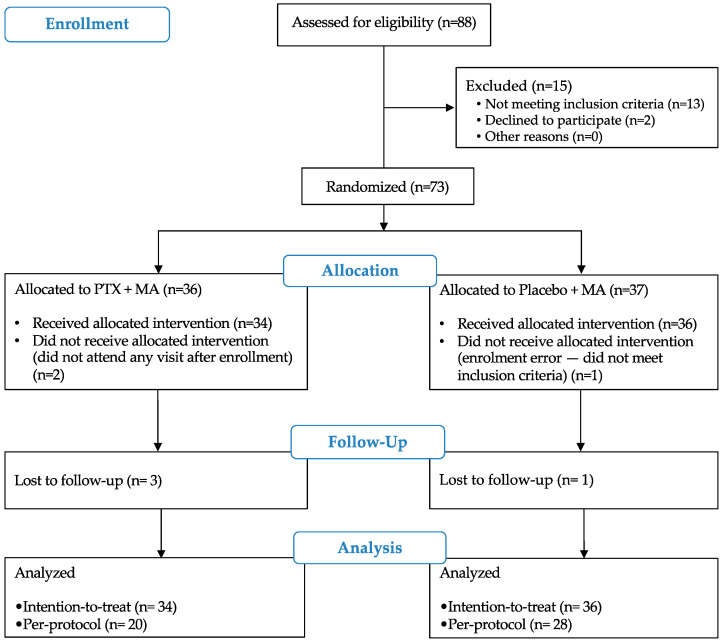
Enrollment and follow-up of participants. MA: meglumine antimoniate; PTX: pentoxifylline.

**Figure 2 pathogens-11-00378-f002:**
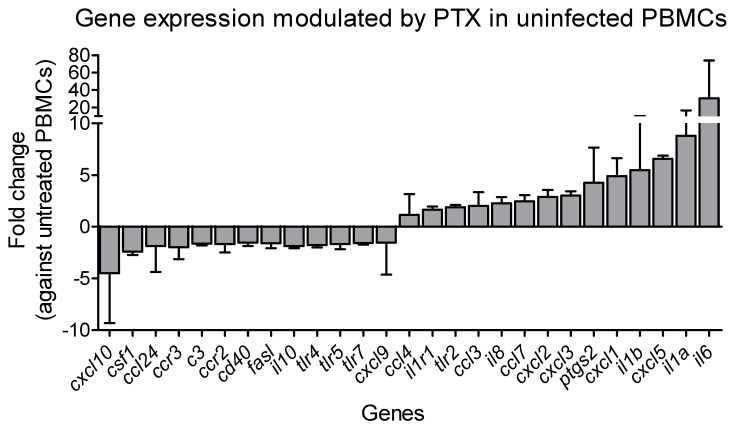
Expression of mediator and receptor genes modulated in uninfected PBMCs (peripheral blood mononuclear cells) after ex vivo exposure to PTX. Gene expression of 27 significantly modulated mediators and receptors in PBMCs from three CL patients exposed during 24 h to 200 µM PTX. Data are expressed as fold regulation (fold change > 2) of PTX exposed PBMCs compared to untreated PBMCs from the same donor (for 2^−ΔΔCt^ values < 1.0, the −1/(2^−ΔΔCt^) is plotted). Genes with a Benjamini–Hochberg (BH) multiple-testing adjusted *p* value of <0.05 were defined as differentially expressed.

**Figure 3 pathogens-11-00378-f003:**
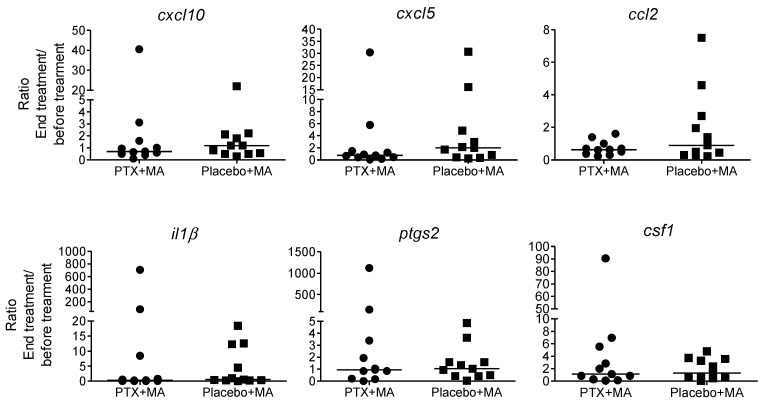
Comparison of expression of inflammatory genes *cxcl10*, *cxcl5* and *ccl2*, *il1b*, *ptgs2* and *cfs1* in patients treated in vivo with MA in combination with PTX or placebo. PBMCs obtained from patients pre-treatment and post-treatment (day 20), were exposed to ex vivo infection with *L. (V.) panamensis*. Expression of these mediators was evaluated by RT-qPCR. A Mann–Whitney test was used for the comparison of the medians of each study group (*n* = 11 per group). Values represent the ratio of the fold change value of the post-treatment visit and the value of fold change of the pre-treatment visit.

**Figure 4 pathogens-11-00378-f004:**
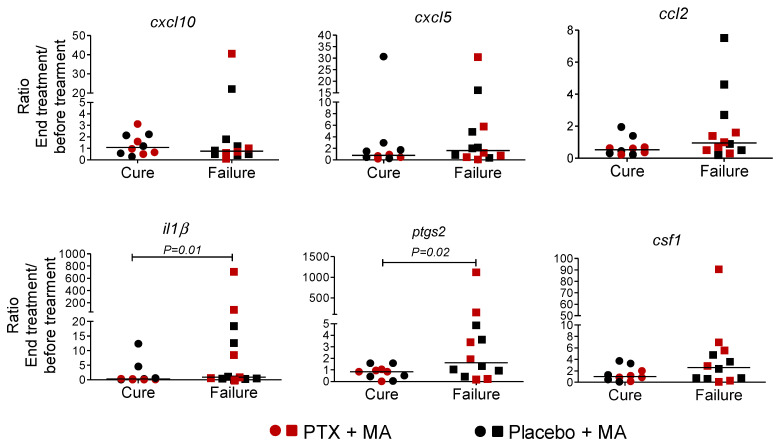
Comparison of expression of inflammatory genes *cxcl10*, *cxcl5* and *ccl2*, *il1b*, *ptgs2* and *cfs1* in PBMCs from patients presenting cure and failure of treatment with MA in combination with PTX or placebo. PBMCs obtained from patients pre-treatment and post-treatment (day 20) were exposed to ex vivo infection with *L. (V.) panamensis*. Expression of these mediators was evaluated by RT-qPCR. A Mann–Whitney test was used for the comparison of the medians of patients with cure (*n* = 10) and failure (*n* = 12). Values represent the ratio of the fold change value of the post-treatment visit and the value of fold change of the pre-treatment visit.

**Table 1 pathogens-11-00378-t001:** Clinical and socio-demographic characteristics of study participants.

	PTX + MA (*n* = 34) *n* (%)	Placebo + MA (*n* = 36) *n* (%)
**Demographic Characteristics**		
Male sex	30 (88.24)	34 (94.44)
Ethnicity		
Afro-Colombian	25 (73.53)	26 (72.22)
Indigenous	1 (2.94)	0 (0.00)
Mixed ethnicity	8 (23.53)	10 (27.78)
Age, years; Median (range)	28 (18–60)	36 (18–62)
**Clinical characteristics**		
Weight (Kilograms); Mean (SD)	65.73 (8.49)	67.73 (7.82)
History of previous leishmaniasis	0 (0)	0 (0)
Number of lesions		
1–3	25 (73.53)	32 (88.89)
>3	9 (26.47)	4 (11.11)
Duration of oldest lesion (months); Median (range)	1 (1–12.5)	2 (1–6)
Maximum diameter of lesions (centimeters)		
<5	21 (61.76)	25 (69.44)
≥5	13 (38.24)	11 (30.56)
Type of lesions *		
Ulcer	32 (94.12)	33 (91.67)
Non-ulcerated lesion	2 (5.88)	3 (8.33)
Presence of lymphadenopathy	7 (20.59)	7 (19.44)
Regional lymphadenopathy	7 (20.59)	12 (33.33)
*Leishmania* species		
*L. (V.) panamensis*	27 (79.41)	23 (63.89)
*L. (V.) guyanensis*	1 (2.94)	0 (0.00)
*L. (V.) braziliensis*	1 (2.94)	3 (8.33)
Not isolated	5 (14.71)	10 (27.78)
Location of lesions, by number of lesions	*n* = 82	*n* = 70
Head or neck	12 (14.63)	12 (17.14)
Arms	45 (54.88)	35 (50.0)
Trunk	9 (10.98)	4 (5.71)
Legs	16 (19.51)	19 (27.14)

* Defined as patients with at least one ulcerated lesion. SD: Standard deviation.

**Table 2 pathogens-11-00378-t002:** Response to treatment per trial arm.

	Treatment		OR *	CI (95%)	*p*
Therapeutic Response	PTX + MA	Placebo + MA			
	*n* (%)	*n* (%)			
Intention-to-treat (*n* = 70)					
Cure	22 (64.71)	27 (75.00)	0.61	0.21–1.71	0.35
Failure	12 (35.29)	9 (25.00)			
Per-protocol (*n* = 48)					
Cure	14 (70.00)	23 (82.14)	0.50	0.13–1.97	0.33
Failure	6 (30.00)	5 (17.86)			

* OR: Odds Ratio.

**Table 3 pathogens-11-00378-t003:** Frequency and intensity of observed clinical and laboratory adverse events (AEs) per treatment arm.

	PTX + MA (*n* = 34) *n* (%)	Placebo + MA (*n* = 36) *n* (%)	*p*
Clinical Adverse Events			
Fever			
Grade 1	14 (41.18)	17 (47.22)	0.61 *
Grade 2	2 (5.88)	0 (0.00)	0.23 **
Headache			
Grade 1	12 (35.29)	10 (27.78)	0.50 *
Grade 2	3 (8.82)	1 (2.78)	0.35 **
Arthralgia			
Grade 1	10 (29.41)	10 (27.78)	0.88 *
Grade 2	0 (0.00)	1 (2.78)	1.00 **
Injection site reaction			
Grade 1	8 (23.53)	8 (22.22)	0.90 *
Grade 2	1 (2.94)	1 (2.78)	1.00 **
Myalgia			
Grade 1	6 (17.65)	5 (13.89)	0.66 *
Grade 2	1 (2.94)	1 (2.78)	1.00 **
Dizziness			
Grade 1	8 (23.53)	3 (8.33)	0.08 *
Laboratory Adverse Events			
Increased amylase			
Grade 1: > ULN - 1.5 × ULN	9 (26.47)	10 (27.78)	0.90 *
Grade 2: > 1.5 - 2.0 × ULN	2 (5.88)	1 (2.78)	0.61 **
Grade 3: > 2.0 - 5.0 × ULN	0 (0.00)	1 (2.78)	1.00 **
Anemia			
Hemoglobin level. Grade 1: < ULN - 10.0 g/dL	10 (29.41)	9 (25.00)	0.68 *
Increased Alanine aminotransferase			
Grade 1: > ULN - 3.0 × ULN	7 (20.59)	9 (25.00)	0.66 *
Grade 2: > 3.0 - 5.0 × ULN	1 (2.94)	0 (0.00)	0.49 **
Grade 3: > 5.0 - 20.0 × ULN	1 (2.94)	0 (0.00)	0.48 **
Increased Aspartate aminotransferase			
Grade 1: > ULN - 3.0 × ULN	7 (20.59)	10 (27.78)	0.48 *
Grade 2: > 3.0 - 5.0 × ULN	1 (2.94)	0 (0.00)	
Other adverse events ***			
Grade 1	20 (58.82)	25 (69.44)	0.35 *
Grade 2	4 (29.79)	4 (11.11)	1.00 **
Grade 3	1 (2.94)	0 (0.00)	0.48 **

* Χ^2^ test, ** Fisher’s exact test, *** e.g., abdominal pain, anorexia, etc. ULN: upper limit of normal.

## Data Availability

The data presented in this study are available on request, subject to authorization by the CIDEIM ethics committee. Requests can be sent to the institutional Ethics Committee of CIDEIM at: cideim@cideim.org.co. The data are not publicly available due to some of the variables being considered sensitive under Colombian regulations. We will not seek to impose restrictions on the purposes for which the data may be used. However, we will require the following: (1) the applicant(s) may not pass the data on to any person other than any co-applicants named in the original request to the CIDEIM ethics committee and must implement security measures to prevent such access; (2) the applicant(s) may not seek to identify individuals by data-mining techniques or otherwise.

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
