# Peer review of "Pentoxifylline in the Treatment of Cutaneous Leishmaniasis: A Randomized Clinical Trial in Colombia"

_pathogens, 2022, doi:10.3390/pathogens11030378_

Round 1

Reviewer 1 Report

The use of combination therapies in the treatment of leishmaniasis aims to improve the outcome of available antileishmanial drugs while reducing their adverse effects and toxicity. Previous studies have been used pentoxifylline in combination with pentavalent antimonials to modulate the chronic inflammatory response associated with leishmaniasis showing variable results depending on the clinical presentation of the disease. Even though it is already published that pentoxifylline addition have not therapeutic benefit against CL produced by Leishmania braziliensis, its effect against other species is far less well known. In this work the authors asses pentoxifylline addition to the treatment with pentavalent antimonials, studying its efficacy, safety, and immunomodulatory effect against Leishmania panamensis the most predominant species of cutaneous leishmaniasis (CL) in Colombia. Therefore, this article contributes to knowledge by providing insights for the design of specific therapeutic protocols against CL, involving those drugs that are most effective depending on the Leishmania species involved.

In general, the manuscript is clear and well written, although some English and format errors need to be checked (some double spaces, lack spaces, italics…). Moreover, there are some specific aspects that should be modified:

Specific comments/suggestions for the authors:

Line 79: The authors should define “GCP” before its abbreviation.  

Line 108: The authors should summarize the criteria to consider a patient as “definitive cure” and the day post-treatment in which the analyses were performed, even though it is described in materials and methods.  

Line 109: The authors should reference Table 2 at the end of this sentence.

Line 130: As Table 3 didn’t appear before, this should be table 3 instead of table 4, please correct also in line 141.

Line 181: Statistical analysis employed to assess the differences in gene expression profiles should be specify in figure 2 caption and Figure S1 caption.

Figure S1: The figure is quite difficult to understand, does the fold regulation data represented for each gene correspond to the absolute value when comparing treated and un-treated cells, regardless of whether the gene is down- or up-regulated? Moreover, it is confusing because in the article it is written that ccl2, csf1, cxcl5, il-1b and ptgs2 genes were down- or up-regulated by PTX treatment (lines 186-187) while in Figure S1 these genes apparently were not significantly modulated. Please clarify in the text and figure caption.

Line 190: “The expression of the selected genes was evaluated in PBMCs collected prior to initiation and at end of treatment, in a subset of the participants in the intervention and control arms”. The number of patients samples used for the gene expression analyses should be clarify.

Line 198: Gene’s name should be written in italics.

Line 200: “ex vivo” and “L. panamensis” should be written in italics.

Line 331: HTLV-1 should be defined before its abbreviation

Line 354: Monoclonal antibodies references for leishmania species identification should be specify in material and methods.

Lines 389-401: Commercial manufacturers of reagents used in cell culture (antibiotics, DMSO, RPMI...) must be specified. Some of the referenced manufacturer’s companies include the country but not all of them, please follow the guidelines for the authors of "pathogens" to uniform.

Lines 402-412: Primers used for qPCR genes expression analyses should be listed in materials and methods with their corresponding references.

Conclusions: The authors should include a short description of the conclusion obtained base on their results.

Reviewer 2 Report

Comments:

In this paper investigators have conducted a randomized clinical trial to evaluate the therapeutic effects of immunomodulator pentoxifylline (PTX) in combination with meglumine antimoniate (MA) treatment for cutaneous leishmaniasis. Although the combinational treatment did not show significant therapeutic efficacy compared to the placebo, the study is appropriately conducted. The parameters in the study are properly evaluated and the manuscript is well written. Methods and results are clearly discussed in the manuscript. While this information is in general of interest and very educative, there are several points that have to be addressed by the authors. 

Suggestions/minor comments: 

  • Authors have discussed, decrease in the secretion of proinflammatory cytokines (TNF-α and IFN-γ) without evidence of clinical benefit in the antimony + pentoxifylline group, however it is not clear whether they have studied cytokines secretion by PBMCs at protein levels (No evidence in this paper), or they are referring their previous study or others study? Please provide clear reference (line no. 242). If not, isn’t it worth checking these cytokines at protein level? 
  • The authors should have considered an experiment to measure the capacity of patient macrophages to kill parasites before and after treatment e.g., using luminometric assay of viable parasite burden. 
  • Considering nitric oxide production is one of the main leishmanicidal mechanisms used by macrophages, measurement of nitric oxide synthase, a necessary enzyme for nitric oxide production by the macrophages could be one of the add on parameters to the study. 
  • Manuscript is well written; it is advised that the authors should define abbreviations at least at the first instance. E.g., TNF-α, GCP, IFN-γ, HTLV-1, CTCAE. (Line no. 41, 79, 228, 331, 363 respectively). 
  • Primers used for qPCR genes expression should be listed in the tabular form or as supplementary table. 
  • In materials and methods, specify the monoclonal antibodies references for leishmania species identification.

Round 2
